

# Gaussian state approximation of quantum many-body scars

Wouter Buijsman[1,2]$^\star$ and Yevgeny Bar Lev[1]$^\dagger$

**1** Department of Physics, Ben-Gurion University of the Negev, Beer-Sheva 84105, Israel
**2** Max Planck Institute for the Physics of Complex Systems,
Nöthnitzer Str. 38, 01187 Dresden, Germany

$\star$ buijsman@pks.mpg.de , $\dagger$ ybarlev@bgu.ac.il

## Abstract

Quantum many-body scars are atypical, highly nonthermal eigenstates embedded in a sea of thermal eigenstates that have been observed in, for example, kinetically constrained quantum many-body models. These special eigenstates are characterized by a bipartite entanglement entropy that scales as most logarithmically with the subsystem size. We use numerical optimization techniques to investigate if quantum many-body scars of the experimentally relevant PXP model can be well approximated by Gaussian states. Gaussian states are described by a number of parameters that scales quadratically with system size, thereby having a much lower complexity than generic quantum many-body states, for which this number scales exponentially. We find that while quantum many-body scars can typically be well approximated by (symmetrized) Gaussian states, this is not the case for ergodic (thermal) eigenstates. This observation suggests that the non-ergodic part of the PXP Hamiltonian is related to certain quadratic parent Hamiltonians, thereby hinting on the origin of the quantum many-body scars.

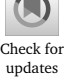

# 1 Introduction

Typical isolated quantum many-body systems thermalize under their own internal dynamics [1–3]. Under time evolution, such systems loose information about their initial condition, leading to the emergence of statistical mechanics. Recent years show a keen interest in quantum many-body systems that fall out of this paradigm. By now, several mechanisms leading to the breakdown of thermalization have been identified, among them many-body localization [4–6], quantum many-body scarring [7–9], and Hilbert space fragmentation [10, 11].

Quantum many-body scarring is a form of ergodicity breaking that can be observed, among others [12], in constrained quantum many-body systems [13]. In contrast to many-body localized systems, quantum many-body scarred systems avoid thermalization only when being initialized in certain highly polarized out-of-equilibrium states [14]. These systems show long-living approximate periodic revivals to their initial state, which can be related to a small number of special, highly nonthermal eigenstates embedded in a sea of thermal eigenstates [15]. These special eigenstates are known as quantum many-body scars, in loose analogy to the single-body quantum scars first observed by Heller in 1984 [16]. Quantum many-body scars have attracted tremendous attention both theoretically [7–9] and experimentally [13, 17–20] in recent years. Several models capturing the phenomenon of quantum many-body scarring have been introduced, with the so-called PXP model for a chain of Rydberg atoms being arguably the most paradigmatic example [14, 15, 21].

The origin of quantum many-body scars is part of a timely debate. For a number of models such as the spin-1 XY model, quantum many-body scarred eigenstates can be constructed analytically [9, 22]. For quantum many-body scars without a known exact form, approximate matrix product states can in certain cases be obtained [23–26]. Other works used for example mean-field like methods to approximate quantum many-body scars [27–29], and it has been suggested that they occur due to proximity of the model to to an integrable point [30–32]. Complementary to the approach used in this work (introduced below), it has also been established that the quantum many-body scars of the PXP model admit a description in terms of polynomially many (in system size) low-lying magnon excitations above the ground state [33].

In this work, we numerically optimize the parameters of the most general non-interacting fermionic system towards a maximum overlap of a (symmetrized) ground state with a given quantum many-body scar of the PXP model. Related optimization procedures for a complementary problem in quantum many-body physics have been found to be fruitful in recent years [34,35]. We find that (symmetrized) Gaussian states typically provide a good description for the quantum many-body scars, indicating that the scars carry certain quadratic features.

The quantum many body scars of the PXP model have a bipartite entanglement entropy that scales at most logarithmically with subsystem size [14, 15]. Since ground states of local quadratic Hamiltontians scale similarly with system size [36], in this work, we investigate if quantum many-body scars of the PXP model can be well approximated by ground states of non-interacting systems. These states belong to the family of Gaussian states (also known as *coherent states*), and are fully described by a number of parameters that scales quadratically with the system size [37], thereby having a much lower degree of complexity than generic quantum many-body states, for which this number scales exponentially. Gaussian states have been found to provide an effective description of many-body eigenstates in a broad range of settings, for example in the context of the mean-field theory of superconductivity [38], or many-body localization [39–46]. Indications for a Gaussian structure of quantum many-body scars would suggest that the non-ergodic part of the PXP Hamiltonian is related to certain quadratic parent Hamiltonians, thereby hinting on the origin of quantum many-body scars.

The outline of this work is as follows. In Section 2, we introduce the model and the properties utilized in the remainder of this work. Section 3 outlines the approximation procedure,

and in Section 4 we present our results. In Section 5, we conclude and outline some possible directions for future investigations.

## 2 Model

We consider the PXP model with periodic boundary conditions,

$$\hat{H}_{\text{PXP}} = \sum_{j=1}^{L} \hat{P}_j \hat{X}_{j+1} \hat{P}_{j+2} \,, \tag{1}$$

where $\hat{X}_j = \hat{\sigma}_j^x$ and $\hat{P}_j = \frac{1}{2}(1 - \hat{\sigma}_j^z)$ with $\hat{\sigma}_j^x$ and $\hat{\sigma}_j^z$ denoting the Pauli $x$ and $z$ operators acting on site $j$, respectively. We impose $\sigma_{L+1}^x \equiv sigma_1^x$ and $\sigma_{L+1}^z \equiv \sigma_1^z$ to account for the periodic boundary conditions. Motivated by experiments for which this model results as an effective description [13], we refer to up and down spin states as "ground" and "excited" states, respectively. We consider the experimentally relevant subspace of the Hilbert space that does not contain two neighboring sites in the excited state. Due to the projector terms $\hat{P}_i$, this subspace is decoupled from the rest of the Hilbert space. Next to the translational symmetry, the Hamiltonian is symmetric with respect to spatial inversion, governed by the operator $\hat{\pi}$, which maps site $i$ to site $L - i + 1$. The Hamiltonian also anticommutes with the parity operator $\hat{\mathcal{C}} = \prod_{i=1}^{L} \hat{\sigma}_i^z$, such that if $|\psi_E\rangle$ is an eigenstate of $\hat{H}_{\text{PXP}}$ with energy $E$, then $\hat{\mathcal{C}}|\psi_E\rangle$ is an eigenstate with energy $-E$. The spectrum is, therefore, symmetric around energy zero and contains an exponentially (in system size) large number of zero-energy eigenstates [47–49]. Since the parity operator has eigenvalues $\pm 1$, any eigenstate of the Hamiltonian can be decomposed as $|\psi_E\rangle = (\hat{P}_+ + \hat{P}_-)|\psi_E\rangle$, where $\hat{P}_\pm$ are the projectors on the corresponding subspaces of $\hat{\mathcal{C}}$. Applying the operator $\hat{\mathcal{C}}$ to this state gives $\hat{\mathcal{C}}|\psi_E\rangle = (\hat{P}_+ - \hat{P}_-)|\psi_E\rangle$, which, as mentioned above, corresponds to an eigenstate of energy $-E$. Therefore, when $E \neq 0$, $\langle\psi_E|\hat{\mathcal{C}}|\psi_E\rangle = \langle\psi_E|\hat{P}_+|\psi_E\rangle - \langle\psi_E|\hat{P}_-|\psi_E\rangle = 0$, and we see that $\langle\psi_E|\hat{P}_+|\psi_E\rangle = \langle\psi_E|\hat{P}_-|\psi_E\rangle = 1/2$.

Since we aim to compare eigenstates of a spin model with those of a fermionic model, we express the Hamiltonian of the PXP model in terms of fermionic operators through a Jordan-Wigner transformation,

$$\hat{c}_j^\dagger = e^{i\pi \sum_{k=1}^{j-1} \hat{P}_k} \hat{\sigma}_j^+ \,, \tag{2}$$

$$\hat{c}_j = e^{-i\pi \sum_{k=1}^{j-1} \hat{P}_k} \hat{\sigma}_j^- \,, \tag{3}$$

with $\hat{\sigma}_j^x = \frac{1}{2}(\hat{\sigma}_j^+ + \hat{\sigma}_j^-)$. The operators $\hat{c}_j$ and $\hat{c}_j^\dagger$ obey the standard fermionic anti-commutation relations $\{\hat{c}_j, \hat{c}_k\} = \{\hat{c}_j^\dagger, \hat{c}_k^\dagger\} = 0$ and $\{\hat{c}_k, \hat{c}_k^\dagger\} = 1$. It is important to note that the mapping to fermionic operators is not unique, and different mappings can potentially give different results. While the resulting fermionic Hamiltonian is non-local due to uncompensated Jordan-Wigner strings, this is not an issue as we focus on the properties of (quantum many-body scarred) eigenstates instead of properties of the PXP Hamiltonian.

Quantum many-body scars are eigenstates characterized by an anomalously high overlap with the $\mathbb{Z}_2$-ordered states $|\mathbb{Z}_2\rangle = |\bullet\circ\bullet\circ\cdots\bullet\circ\rangle$ and $|\mathbb{Z}_2'\rangle = |\circ\bullet\circ\bullet\cdots\circ\bullet\rangle$, where $\circ$ and $\bullet$ are pictorial representations of a site in the ground and excited state, respectively [14,15]. These special eigenstates display a bipartite entanglement entropy that scales at most logarithmically with subsystem size. The null space of the Hamiltonian is also known to host a quantum many-body scar for certain system sizes [14,23,50]. As motivated below, we do not consider these zero-energy scars in this work. The quantum-many body scars have almost equal energy separations of $\Omega \approx 1.31$, which only weakly depends of the system size. This results in the

appearance of long-lived periodic revivals to the initial state starting from a $\mathbb{Z}_2$-ordered state. The PXP model hosts various additional non-ergodic eigenstates (see, e.g., Ref. [50]), which are sometimes referred to as quantum many-body scars as well. Here, we adapt the more restrictive definition of quantum many-body scars as introduced above. We refer to Ref. [8] for a recent review on the different definitions of quantum many-body scars used in the literature.

## 3 Gaussian state approximation

The most general quadratic Hamiltonian with $L$ fermionic modes is given by

$$\hat{H} = \sum_{j,k=1}^{L} \left[ A_{jk}\hat{c}_j^\dagger \hat{c}_k + \frac{1}{2}\left( B_{jk}\hat{c}_j^\dagger \hat{c}_k^\dagger - B_{jk}^*\hat{c}_j\hat{c}_k \right) \right], \tag{4}$$

where $A$ is Hermitian, $B$ is antisymmetric, and the operators $\hat{c}_j$ and $\hat{c}_j^\dagger$ obey the standard fermionic anticommutation relations as introduced above. Fermions are created and annihilated in pairs, meaning that eigenstates can have either an even or an odd number of fermions. Hamiltonian (4) is diagonalized by a Bogoliubov transformation [37, 51, 52],

$$\hat{d}_j = \sum_k \left( U_{jk}\hat{c}_k + V_{jk}\hat{c}_k^\dagger \right), \tag{5}$$

$$\hat{d}_j^\dagger = \sum_k \left( V_{jk}^*\hat{c}_k^\dagger + U_{jk}^*\hat{c}_k \right), \tag{6}$$

where $U$ and $V$ are required to obey $UV^T + VU^T = 0$ and $UU^\dagger + VV^\dagger = 1$ in order for $\hat{d}_j$, $\hat{d}_k^\dagger$ to obey the fermionic anti-commutation relations. The eigenstates of Hamiltonian (4) are thus given by product states in the basis of the quasi-particles created by $\hat{d}_i^\dagger$ on top of a quasi-particle vacuum.

In this work, we focus on the question whether quantum many-body scars can be well approximated by symmetrized ground state of a non-interacting Hamiltonian,

$$|\psi_\pm\rangle = \mathcal{N}\left( |\psi_0\rangle \pm \hat{\pi}|\psi_0\rangle \right). \tag{7}$$

Here, $\mathcal{N}$ is a normalization factor, and $|\psi_0\rangle$ is the ground-state of Hamiltonian (4). The states are symmetrized to follow the inversion symmetry of the scars, which results in a better approximation (see Appendix A for the overlap of non-symmetrized Gaussian states with quantum many-body scars). We remark that $|\psi_\pm\rangle$ is typically not a Gaussian state itself. For quantum many-body scars which are symmetric with respect to inversion we take $|\psi_+\rangle$, and for quantum many-body scars which are antisymmetric we take $|\psi_-\rangle$.

We look for matrices $A$ and $B$ characterizing the quadratic Hamiltonian (4), which give maximal overlap between $|\psi_\pm\rangle$ and a given quantum many-body scar. To compute the overlap, we take the state $|\psi_\pm\rangle$ in the basis where $\hat{n}_j = \hat{c}_j^\dagger \hat{c}_j$ is diagonal, and the quantum many-body scar in the basis where $\hat{\sigma}_j^z$ is diagonal. The PXP Hamiltonian expressed in this basis is a real-valued matrix. Physically, this means that the time-evolution operator is symmetric under time-inversion. Therefore, we lower the computational costs by restricting $A$ and $B$ to be real. Since the quantum many-body scars have considerable overlap with the $\mathbb{Z}_2$ state, for the initial guess of the matrices $A$ and $B$ we take

$$A = \text{diag}(\pm 1, 1, -1, 1, \ldots, -1, 1), \qquad B = 0, \tag{8}$$

such that the initial ground state of (4) is given by the $\mathbb{Z}_2$ state. The number of fermions in this ground state corresponds to the number of $(-1)$'s on the main diagonal of $A$. Since in the

fermionic language the parity operator is given by $\hat{\mathcal{C}} = (-1)^{\hat{N}}$, where $\hat{N}$ is the operator counting the number of fermions, a state with an even (odd) number of fermions is an eigenstate of the parity operator with eigenvalue $+1$ ($-1$). By changing the sign of the (arbitrarily chosen) first element on the main diagonal of the initial guess for $A$ we can control the evenness of the fermion number and as such the parity of the ground state. Taking the diagonal elements of $A$ as $\pm 1$ in a randomized fashion, such that the initial guess of the ground state is given by a randomly chosen product state, leads to significantly lower optimized overlaps.

For the optimization procedure we use the Limited-Memory Broyden-Fletcher-Goldfarb-Shanno (also known as LM-BFGS) algorithm [53], which we terminate when the gradient of the overlap with respect to the optimization parameters is equal to zero up to numerical precision. In short, this algorithm works in two steps. First, it determines the Hessian of the cost function (here, the overlap) in order to determine the direction in which the increase is maximal. Second, it optimizes the step size in this direction such that the increase is at its maximum. For small system sizes, we have tested the performance of all optimization algorithms implemented in the Python SciPy package [54]. We empirically observed that this algorithm provides optimal results in terms of the optimized overlaps. We remark that it is generically impossible to analytically find the optimal parameters of a Gaussian state approximating a given many-body state [55]. As the optimization algorithm (like all numerical optimization algorithms) searches for local maxima, it is important to ensure the initial guess to be as close as possible to the desired result. One could alternatively choose the initial guesses of $A$ and $B$ such that the ground state of $\hat{H}$ is given by the product state that has the highest overlap with the quantum many-body scar under consideration. Typically, but not always, this is the $\mathbb{Z}_2$ state. For the quantum many-body scars closer to the center of the spectrum, this is not always the $\mathbb{Z}_2$ state. For this, we find qualitatively similar results. Initializing $A$ and $B$ with random elements (subject to the symmetry constraints) gives, as expected, very low optimized overlaps.

We consider relatively modest system sizes due to the high computational costs of the optimization procedure. Typically, optimization requires several thousands, or with outliers, several tens of thousands evaluations of the overlap. For each such overlap the *many-body* ground-state of the quadratic system has to be re-calculated. We note that, while the single-particle states of a quadratic model can be computed in time polynomially with the system size, the computation of the *many-body* ground state scales exponentially with the system size. We note in passing, that the outlined procedure does *not* typically correspond to the calculation of the natural orbitals from the diagonalization of the single-particle density matrix. In fact, it is known that a ground state constructed in the basis of natural orbitals produces optimal results only for states with two fermions [55,56].

# 4 Results

Since the quadratic Hamiltonian (4) conserves the parity of the number of fermions, we have to approximate the projections of the scars onto different parity sectors, $\hat{P}_{\pm}|\psi_{\mathrm{scar}}\rangle$, separately. as shown in Section 2, all eigenstates of the PXP model (with nonzero energy), including the quantum many-body scars, have the same overlap $1/2$ with both sectors. The projections of the eigenstates on the parity sectors are eigenstates of $\hat{H}_{\mathrm{PXP}}^2$. Because of the particle-hole symmetry of $\hat{H}_{\mathrm{PXP}}$ (the anti-commutation of $H_{\mathrm{PXP}}$ and $\mathcal{C}$), we have that $\hat{H}_{\mathrm{PXP}}^2$ commutes with the parity operator, while $\hat{H}_{\mathrm{PXP}}$ does not. For $\hat{H}_{\mathrm{PXP}}^2$, the number of creation and annihilation operators in each term is even. We note, that for system sizes dividable by 4, the $\mathbb{Z}_2$ state lies in the positive parity sector and in the negative parity sector otherwise.

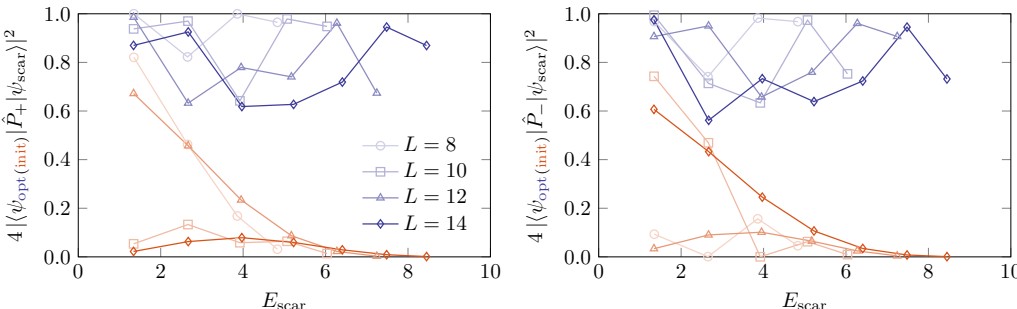

Figure 1: Blue lines (upper sets of curves) show the optimized overlaps $4|\langle\psi_{\text{opt}}|\hat{P}_+|\psi_{\text{scar}}\rangle|^2$ (left panel) and $4|\langle\psi_{\text{opt}}|\hat{P}_-|\psi_{\text{scar}}\rangle|^2$ (right panel) for several system sizes as a function of the energy of the quantum many-body scars. Red lines (lower sets of curves) show the overlap $4|\langle\psi_{\text{init}}|\hat{P}_+|\psi_{\text{scar}}\rangle|^2$ (left panel) and $4|\langle\psi_{\text{init}}|\hat{P}_-|\psi_{\text{scar}}\rangle|^2$ (right panel). The largest possible overlap with this normalization is unity.

By $|\psi_{\text{init}}\rangle$, we denote the symmetrized ground state of Hamiltonian (4) for the initial choice of matrices $A$ and $B$ according to (8). The resulting optimized symmetrized state will be denoted by $|\psi_{\text{opt}}\rangle$. For convenience, in what follows we focus on the quantity $4|\langle\psi_{\text{opt}}|\hat{P}_\pm|\psi_{\text{scar}}\rangle|^2$, which is bounded from above by unity, since $\langle\psi_{\text{scar}}|\hat{P}_\pm|\psi_{\text{scar}}\rangle = 1/2$ (see Section 2). We focus only on scars with positive energies, $E_{\text{scar}} > 0$, since the spectrum of the PXP model is symmetric around zero and the projections on a parity sector of eigenstates with non-zero energy $E$ and $-E$ are identical up to an overall minus sign. We do not consider quantum many-body scars at zero energy, since quantum many-body scars are not uniquely defined due to the numerous degeneracy at zero energy.

Fig. 1 shows the initial $4|\langle\psi_{\text{init}}|\hat{P}_\pm|\psi_{\text{scar}}\rangle|^2$ and optimized overlaps $4|\langle\psi_{\text{opt}}|\hat{P}_\pm|\psi_{\text{scar}}\rangle|^2$ for system sizes $L = 8$ to $L = 14$ as a function of the energies of the scars, $E_{\text{scar}}$. We observe that the optimized overlap is typically close to unity, and shows only a weak system-size dependence. We also note that the optimization leads to a significant improvement of the overlap, as can be seen by comparing to the overlap with the initial guess $|\psi_{\text{init}}\rangle$. As discussed below Eq. (8), this initial guess is given by $|\psi_{\text{init}}\rangle = (|\mathbb{Z}_2\rangle \pm |\mathbb{Z}_2'\rangle)/\sqrt{2}$ when the $\mathbb{Z}_2$ state is in the same parity sector as the optimized state, while the initial guess is the same but with the first spin is flipped when the $\mathbb{Z}_2$ state is not in the same parity sector. In Appendix B, we show that significantly lower overlaps are obtained for non-scarred (thermal) eigenstates, in particular at larger system sizes or when the $\mathbb{Z}_2$ state is not in the same parity sector as the optimized state. We note in passing, that although the optimized states are not confined to the constrained Hilbert space of the PXP model by construction, the high overlap implies that the optimized states almost fully reside there.

Quantum many-body scars distinguish themselves from other types of non-ergodic many-body states by their anomalously high overlap with the $|\mathbb{Z}_2\rangle$ and $|\mathbb{Z}_2'\rangle$ states. This can be seen at the lower (red) set of lines in Fig. 1, which shows the overlap, $4|\langle\psi_{\text{init}}|\hat{P}_+|\psi_{\text{scar}}\rangle|^2$, where, as mentioned before, $|\psi_{\text{init}}\rangle = (|\mathbb{Z}_2\rangle \pm |\mathbb{Z}_2'\rangle)/\sqrt{2}$ in the left panel when $L$ is dividable by 4 and in the right panel otherwise. In Fig. 2, we see that also the optimized state has a qualitatively similar overlap with the $|\mathbb{Z}_2\rangle$ and $|\mathbb{Z}_2'\rangle$ states, by plotting $|\langle\psi_{\text{opt}}|\mathbb{Z}_2\rangle|^2$ as a function of the energy of the quantum many-body scars for the system sizes considered above. It is interesting to note that at the edges of the spectrum, the optimized and initial states are almost orthogonal to each other.

The structure of the optimized matrices A and B could provide insight on the structure of the quantum many-body scars. Fig. 3 shows color plots of the optimized matrices for the scar with the second-highest (the highest-energy scar is the ground state) energy at system

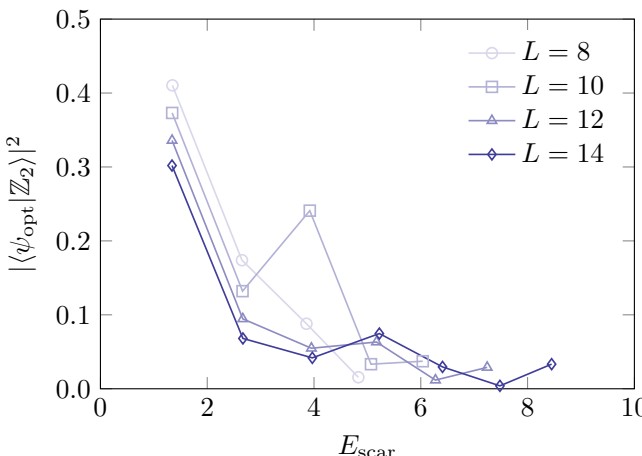

Figure 2: The overlap $|\langle\psi_{\mathrm{opt}}|\mathbb{Z}_2\rangle|^2$ as a function of the energy of the quantum many-body scars at several system sizes.

size $L = 14$. The Jordan-Wigner transformation (2)-(3) is rooted at the first site, thereby breaking the translational invariance of the resulting fermionic Hamiltonian. This is reflected in the matrices $A$ and $B$ showing alternating positive and negative elements. The optimized Hamiltonian exhibits a notion of locality, reflected in the band-like structure. The quantum many-body scars are known to have decaying spatial correlations and sub-volume law scaling of the entanglement entropy. It is thus natural that the optimal quadratic Hamiltonians are local. With this said, we do not constrain the approximating quadratic Hamiltonians, such that the outcome of the optimization could have been non-local. We observe a checkerboard-pattern of the matrices $A$ and $B$ that suggests a translational invariance for translations over two sites. We note that the symmetrization of a state invariant by translation over two sites leads to a state translationally invariant by one site. Then, the trial wavefunction is almost translationally invariant. In Appendix A, we show that significantly lower overlaps are obtained when the trial wavefunction is not symmetrized. This provides a hint that the (non-Gaussian) quantum many-body scars are close to superpositions of two Gaussian states invariant under translations by two sites. Although (non-symmetrized) Gaussian states can be translationally invariant, our results suggests that such states are not sufficient to capture the structure of the (non-Gaussian) quantum many-body scars properly.

## 5 Conclusions and outlook

Quantum many-body scars are states with low entanglement embedded in ergodic (thermal) eigenstates. In this work, we have studied to what extent quantum many-body scars in the PXP model can be described by inversion symmetrized Gaussian states, corresponding to a symmetrized ground state of a quadratic Hamiltonian with no particle number conservation. It is not guaranteed a priori that quantum many-body scars can be described by such states since not all low-entangled states are Gaussian. We numerically optimized the parameters of the most general quadratic fermionic Hamiltonian such that the (symmetrized) ground state has maximal overlap with the quantum many-body scar under consideration. We found that quantum many-body scars can typically be well described by states of this form. We also showed that the optimal quadratic Hamiltonian is local, has a non-negligible pairing, and is translationally invariant only every *two* sites. Significantly lower overlaps are obtained when the trial wavefunction is not symmetrized. Since entanglement entropy of ground-states of

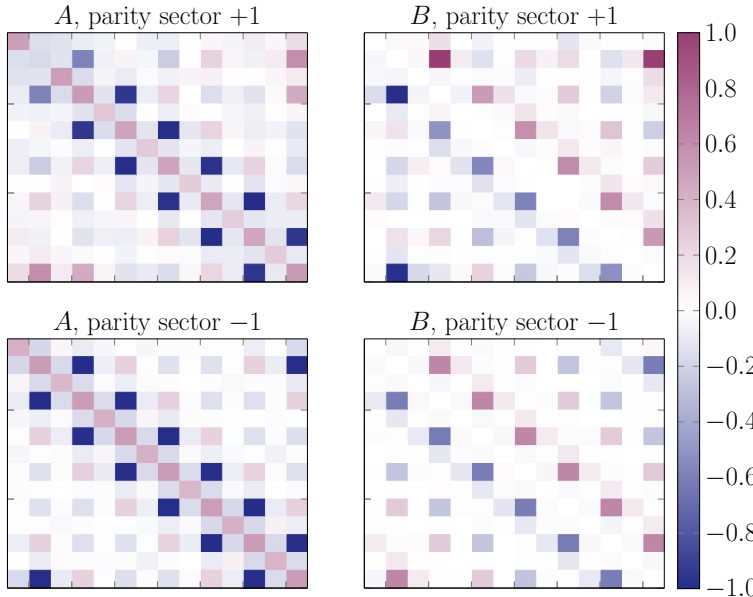

Figure 3: Color plots of the matrices $A$ (left) and $B$ (right) of the quadratic Hamiltonian (4), whose ground-state has the largest overlap with the scar with the second-highest energy for $L = 14$. The top panels correspond to optimization with respect to $\hat{P}_+|\psi_{\text{scar}}\rangle$ and the bottom panels with respect to $\hat{P}_-|\psi_{\text{scar}}\rangle$. The scale has been chosen such that the largest absolute value is unity.

local quadratic Hamiltonians scales logarithmically with the system size [36], our result suggests that a similar scaling will hold also for quantum many-body scars. It is important to note that the optimization of ergodic, (thermal) eigenstates leads to very poor results, indicating that while the scarred states have Gaussian structure, this is not the case for the ergodic states. In the literature, various other approximations of quantum many-body scars have been suggested. Possibly (in spirit) closest to our approach are the approximations by bosonic quasi-particle excitations on top of certain reference states [33], or these following from mean-field like approaches [27]. Although these approaches yield (slightly) better overlaps, we believe that the quality of our results does not rule out an underlying structure of quantum many-body scars in terms of Gaussian states.

In this work, we have used a distinct quadratic Hamiltonian for each quantum many-body scar. In future studies, it would be interesting to see if a single optimal quadratic Hamiltonian can be used to capture the structure of each of the scars, and also to understand the origin of such an effective single-particle description. One might hypothesize that the single-particle eigenstates of the quadratic Hamiltonians become (almost) identical in the thermodynamic limit. Alternatively, one might speculate that the PXP Hamiltonian has some hidden block-diagonal structure with the quantum many-body scars given by the ground state of the blocks. Then, no single quadratic Hamiltonian is expected to exist. A related open question is whether multiple distinct parent Hamiltonians can be found for a given quantum many-body scar. This question is relevant, in particular, when aiming to unify the various approximation schemes that have been proposed in the literature. It would be also interesting to investigate further if similar results can be obtained for other quantum many-body scarred models.



## Acknowledgements

**Funding information**    This research was supported by a grant from the United States-Israel Binational Foundation (BSF, Grant No. 2019644), Jerusalem, Israel, and the United States National Science Foundation (NSF, Grant No. DMR-1936006), and by the Israel Science Foundation (grants No. 527/19 and 218/19). W.B. acknowledges support from the Kreitman School of Advanced Graduate Studies at Ben-Gurion University.

## A    Optimization results for non-symmetrized Gaussian states

Here, we study the overlap of quantum many-body scars with non-symmetrized optimized Gaussian states, here denoted by $|\psi_{\mathrm{opt},0}\rangle$, instead of the symmetrized version $|\psi_{\mathrm{opt}}\rangle$ [see (7)]. As this investigation is only for illustrative purposes, we restrict the analysis to optimization with respect to the parity sector containing the $\mathbb{Z}_2$ state, which is arguably physically the most interesting. Fig. 4 shows the optimized overlaps as a function of the energy of the quantum many-body scar for several system sizes. Comparing the results with those shown in Fig. 1, we observe a substantially lower overlap, which highlights the importance of symmetrization.

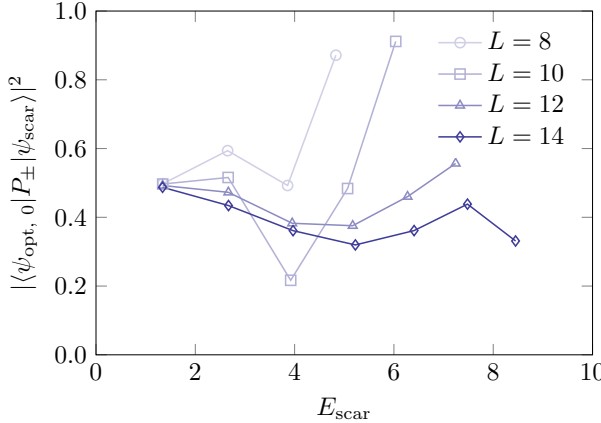

Figure 4: The optimized overlaps $4|\langle\psi_{\mathrm{scar}}|\hat{P}_{\pm}|\psi_{\mathrm{opt},0}\rangle|^2$ for several system sizes as a function of the energy of the quantum many-body scars. The sign of the projector $P_{\pm}$ is chosen such that it projects on to the parity sector containing the $\mathbb{Z}_2$ state. The largest possible overlap with this normalization is unity.

## B    Optimization results for non-ergodic eigenstates

Here, we study the overlap of non-ergodic (thermal) eigenstates with optimized (symmetrized) Gaussian states. We restrict the analysis to optimization with respect to the parity sector containing the $\mathbb{Z}_2$ state, which is arguably physically the most interesting. For each system size, we consider the eigenstates with energies closest to integers. If such an eigenstatate is a quantum many-body scar, we take the eigenstate which is second-closest in energy. Fig. 5 shows the results. The overlaps for thermal eigenstates are significantly lower than those for quantum many-body scars (cf. Fig. 1). We thus see that our approach works well for quantum many-body scars, while it works significantly less well for thermal eigenstates.

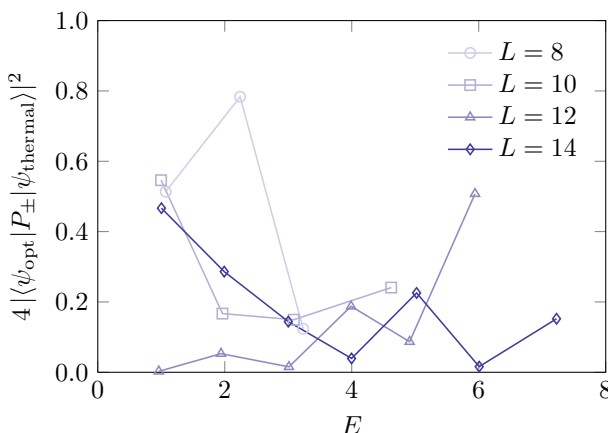

Figure 5: The optimized overlaps $4|\langle\psi_{\text{scar}}|\hat{P}_{\pm}|\psi_{\text{opt}}\rangle|^2$ for several system sizes as a function of the energy for thermal eigenstates. See the main text for the choice of the eigenstates. The sign of the projector $P_{\pm}$ is chosen such that it projects on to the parity sector containing the $\mathbb{Z}_2$ state. The largest possible overlap with this normalization is unity.

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
