# Peer review of "Gaussian state approximation of quantum many-body scars"

_SciPost Physics, doi:SciPost Phys. 17, 055 (2024)_

## Round 1 · Referee Report · Anonymous (Referee 1) · 2023-8-19

Strengths

Motivated by the empirically known low bipartite entanglement entropy of atypical eigenstates of the PXP model, the authors try to approximate these states by ground state wavefunctions of a quadratic, fermionic Bogoliubov-de Gennes (BdG)-like Hamiltonian.
The authors optimize the overlap between the scar states and the ansatz wavefunctions by tuning the coefficients of the BdG Hamiltonian, in particular the hopping matrix. This numerical study furnishes the following two main results:
1. the trial wavefunctions have appreciable overlap with the exact eigenstates, whereby it is higher for the scar states at the edges of the spectrum;
2. the optimal hopping matrix seems to be local, which gives back a low entanglement of the scar states.

Weaknesses

Unfortunately, it remains unclear what one should take home from this analysis. It seems that the trial wavefunctions are rather tedious to obtain, and contain many more parameters than the wavefunctions of Ref. [25-27]. The authors do not benchmark their approach with the existing approaches: Are their wavefunctions more or less accurate, and does the gained accuracy (if any) justify the computational expense to optimize the wavefunctions? What is the physical content conveyed by the ansatz (if any)? What more does one learn as compared to what one learns from the wavefunctions proposed and discussed in the literature?

At a technical level, the approximation is not well described: A bosonic wavefunction is approximated by a fermionic wavefunction. While the Hilbert spaces are of equal dimension, there are many ways to map states with the same site occupations from fermions to bosons. It should be specified which mapping (ordering of fermionic operators) was chosen and why. It seems very likely that different choices of orderings would yield different results. Is there an optimal choice?

Report

Unfortunately, neither of the 4 possible acceptance criteria is met.

Requested changes

1. Considering that the ground state of PXP, as a ground state of a gapped one-dimensional Hamiltonian, is expected to be low-entangled, property (i) seems not so surprising, and is in fact shared with several other trial wavefunctions in the literature.
Please comment.

2. To what extent the optimal hopping is really short range is hard to tell, since the authors do not provide data for different sizes. The resulting hopping range is still a quarter to a third of the system size. How does this range evolve with system size? Does it seem to saturate? It probably should, since otherwise the state would hardly have a low entanglement entropy, unlike the state to be approximated. Thus, the short range of hoppings seems to be rather a sanity check of the approximation than an actual result that bears physical content.
Since the original PXP model has open boundary and thus is not translation invariant, one should not expect translation invariant hoppings to emerge of course. However, at least in the bulk one would expect translation invariance to be recovered, as seems to happen, looking at Fig.3; however the authors claim otherwise. This should be elaborated on more if the authors claim absence of translation invariance. If they believe it is there: why should this happen, and what is its significance?

3. The first sentences of the abstract and in the second paragraph are rather misleading, as they give the impression that quantum many-body scars are only found in kinetically constrained models. However, to our knowledge there are more models with many body scars in unconstrained models (Hubbard models, XY models, Heisenberg models etc...) than in kinetically constrained models. Please reformulate.

4. Can the authors motivate their choice of open boundary conditions (even though periodic boundary conditions would significantly reduce the Hilbert space)? Presumably this is due to a problem for the mapping between fermionic and bosonic wavefunctions. If so, that should be spelt out.

5. It is unclear what is meant by "the PXP model is time-reversal symmetric" before Eq.(6). Is its meaning that the Hamiltonian has real matrix elements in the sz basis (but not standard TR-invariance of spin systems)?

6. On page 3 it is stated that many-body scars distinguish themselves from other types of non-ergodic many-body states by their overlap with Neel states. This is a rather confusing statement. Usually one considers all non-ergodic many-body states in a given model as scar states.

  • validity: low
  • significance: poor
  • originality: low
  • clarity: ok
  • formatting: reasonable
  • grammar: acceptable

Author:  Wouter Buijsman  on 2024-01-08  [id 4225]

(in reply to Report 1 on 2023-08-19)

We would like to thank the Referee for their careful reading of the manuscript, as well as their useful comments and suggestions. Please find below our replies.

The Referee writes:

Unfortunately, it remains unclear what one should take home from this analysis. It seems that the trial wavefunctions are rather tedious to obtain, and contain many more parameters than the wavefunctions of Ref. [25-27]. The authors do not benchmark their approach with the existing approaches: Are their wavefunctions more or less accurate, and does the gained accuracy (if any) justify the computational expense to optimize the wavefunctions? What is the physical content conveyed by the ansatz (if any)? What more does one learn as compared to what one learns from the wavefunctions proposed and discussed in the literature?

Our response: The goal of our work is to show that the quantum many-body scars of the PXP model can be well-approximated by Gaussian states. We believe that this considerably advances the understanding of the origin of quantum many-body scars, since one could argue that the PXP model might be in the vicinity of certain quadratic parent Hamiltonians. We did not intend to develop an efficient method that minimizes the number of optimization parameters, or provides a better overlap compared to existing methods. Our motivation is thus different from other works, for example focusing on matrix product states, where in some cases exact matrix product state constructions of quantum many-body scars can be found. We have clarified our motivation in the Abstract and Introduction of the revised version.

The Referee writes:

At a technical level, the approximation is not well described: A bosonic wavefunction is approximated by a fermionic wavefunction. While the Hilbert spaces are of equal dimension, there are many ways to map states with the same site occupations from fermions to bosons. It should be specified which mapping (ordering of fermionic operators) was chosen and why. It seems very likely that different choices of orderings would yield different results. Is there an optimal choice?

Our response: We thank the Referee for pointing this out. In the revised version, we have added more details on the mapping of the PXP model into fermionic form. It is an open question what is the optimal mapping.

The Referee writes:

Considering that the ground state of PXP, as a ground state of a gapped one-dimensional Hamiltonian, is expected to be low-entangled, property (i) seems not so surprising, and is in fact shared with several other trial wavefunctions in the literature. Please comment.

(i) The trial wavefunctions have appreciable overlap with the exact eigenstates, whereby it is higher for the scar states at the edges of the spectrum.

Our response The main point of our work is that Gaussian states can provide surprisingly good descriptions of the quantum many-body scars of the PXP model, compared to any other eigenstates of the same model. This is surprising, since the model is interacting, and the approximation works amazingly well also in the middle of the spectrum. We would like to stress that the fact that the ground state of the PXP model is low entangled does not guarantee it is well approximated by a Gaussian state. For example, a cat state is a superposition of two Gaussian states. It has area-law entanglement, but a maximal overlap of $1/2$ with a Gaussian state. The fact that our approach works better for the ground state is a side point.

While as we mention in the Introduction, the fact that the quantum many-body scars have low entanglement motivated us to conduct this study, there is no a-priori reason to believe that Gaussian states would well-approximate the quantum many-body scars (or the ground state). We have commented on this in the revised version.

The Referee writes:

To what extent the optimal hopping is really short range is hard to tell, since the authors do not provide data for different sizes. The resulting hopping range is still a quarter to a third of the system size. How does this range evolve with system size? Does it seem to saturate? It probably should, since otherwise the state would hardly have a low entanglement entropy, unlike the state to be approximated. Thus, the short range of hoppings seems to be rather a sanity check of the approximation than an actual result that bears physical content.

Our response: The Referee is right that once we show that the Guassian states provides good descriptions of the quantum many-body scars, the fact that the matrices $A$ and $B$ have large elements only close to the diagonal is merely a sanity check. To be slightly more quantative, we bring here (see the Figure in the attachment) a plot of the cut throught the anti-diagonal of the matrices depicted in Fig. 3. The decay of the hopping when moving away from the diagonal is visible. The main physical content, which should be infered from Fig. 3 is (a) the pairing matrix is not-negligible, and (b) translation invariance is broken, even in the bulk (more about this below).

The Referee writes:

Since the original PXP model has open boundary and thus is not translation invariant, one should not expect translation invariant hoppings to emerge of course. However, at least in the bulk one would expect translation invariance to be recovered, as seems to happen, looking at Fig. 3; however the authors claim otherwise. This should be elaborated on more if the authors claim absence of translation invariance. If they believe it is there: why should this happen, and what is its significance?

Our response: We thank the Referee for pointing this out. The claim on the absence of translational invariance refers to the checkerboard-pattern of the matrices $A$ and $B$ that suggests a translational invariance for translations over two sites, instead of ``full'' translational invariance. This does not appear to be surprising given the fact that our trail wavefunctions [Eq. (5)] break translational invariance of the matrices $A$ and $B$. We found that enforcing translation invariance in the trial wavefunctions yields considerably lower overlaps. This provides a hint that the exact quantum many-body scars are superpositions of states invariant under translations by two sites. We have added an elaboration on this in the revised version.

The Referee writes:

The first sentences of the abstract and in the second paragraph are rather misleading, as they give the impression that quantum many-body scars are only found in kinetically constrained models. However, to our knowledge there are more models with many body scars in unconstrained models (Hubbard models, XY models, Heisenberg models etc...) than in kinetically constrained models. Please reformulate.

Our response: We thank the Referee for pointing this out. We have corrected this in the revised version.

The Referee writes:

Can the authors motivate their choice of open boundary conditions (even though periodic boundary conditions would significantly reduce the Hilbert space)? Presumably this is due to a problem for the mapping between fermionic and bosonic wavefunctions. If so, that should be spelt out.

Our response: We use open boundary conditions to avoid the extra boundary term, which appears after mapping the spins into fermions using a Jordan-Wigner transformation.

The Referee writes:

It is unclear what is meant by "the PXP model is time-reversal symmetric" before Eq. (6). Is its meaning that the Hamiltonian has real matrix elements in the $S_z$-basis (but not standard translational-invariance of spin systems)?

Our response: Yes, practically this means that the PXP Hamiltonian is real-valued in the $S_z$-basis. We have elucidated this in the revised version.

The Referee writes:

On page 3 it is stated that many-body scars distinguish themselves from other types of non-ergodic many-body states by their overlap with N\' eel states. This is a rather confusing statement. Usually one considers all non-ergodic many-body states in a given model as scar states.

Our response: Although certain works (Ref. [46], for example) classify all non-ergodic eigenstates as quantum many-body scars, this is in our view not a standard convention. See Ref. [8], Sec. VI. C for a recent overview of the debate on this issue. We have elaborated on our precise definition of quantum many-body scars in the revised version.

Attachment:

attachment.pdf

---

## Round 1 · Referee Report · Anonymous (Referee 2) · 2023-10-16

Strengths

1. This is an interesting question to explore.

2. The results are presented clearly and are interesting.

Weaknesses

1. The analysis seems shallow, and many immediate questions and issues are not addressed.

2. Does not provide adequate review of the algorithm and methodology used.

Report

This manuscript contains results on some properties of Quantum Many-Body Scars (QMBS) of the PXP model. The authors explore if these QMBS eigenstates can be approximated by the ground state of some non-interacting Bogoliubov-deGennes (BdG) fermion model, which is not necessarily local or translation-invariant. They optimize the parameters in the fermion model to maximize its overlap with the QMBS, and they use a previously developed algorithm for this optimization. They show that the overlap is large for QMBS at the edge of the spectrum of the PXP model, it is lesser in the middle of the spectrum. Further, for the QMBS, the optimized model has some locality that emerges even though it is not assumed from the start.

Given that ground states of non-interacting fermion states have low-entanglement and so do QMBS eigenstates of the PXP model, trying to explore if there is any connection between them is an interesting question. However, I think the current version of the manuscript does not present a thorough enough exploration of this question to warrant immediate publication in SciPost. Here are some questions that the authors could consider exploring.

1. The authors present the obtained results for the QMBS as-is, without any exploration on how surprising their results actually are. For example, is it clear that the overlap of the QMBS with optimized Gaussian states is significantly different from that of other eigenstates in the spectrum? If not, is there some properties of the optimized Hamiltonian that change for non-QMBS?

2. The ground state of the PXP model and its QMBS close to the edges of the spectrum are known to be well-approximated by Matrix Product States: https://arxiv.org/abs/1903.10517. This property itself makes it plausible that the QMBS states can admit several approximate parent Hamiltonians. Is this property related to the optimized non-interacting Hamiltonians that the authors discover for the QMBS close to the edge of the spectrum? For example, do the exact MPS studied in the above paper admit such non-interacting parent Hamiltonians?

3. The algorithm for optimization is used as a ``black box" and no details or motivations are presented. For example, what is the role of the initial states $|\psi_{init}\rangle$ that dictates the initial guesses for the matrix A and B? Do answers change for different choices of this initial state?

4. Another question that the authors may consider exploring if computationally feasible: Since the ground state of the free-fermion ground state does not naturally contain the Rydberg blockade constraint, do the overlaps with the QMBS change if the many-body wavefunctions are projected and normalized within the constrained subspace?

Requested changes

1. The results on QMBS eigenstates should be contrasted with results for non-QMBS eigenstates. Perhaps the same algorithms can be applied to non-QMBS, and properties that distinguish the QMBS can be pointed out clearly.

2. Potential connections to previous work addressing similar questions (https://arxiv.org/abs/1903.10517) should be discussed clearly.

3. Would be good to have an overview of important details of the algorithm used.

  • validity: high
  • significance: good
  • originality: ok
  • clarity: good
  • formatting: excellent
  • grammar: excellent

Author:  Wouter Buijsman  on 2024-01-08  [id 4226]

(in reply to Report 2 on 2023-10-16)

Let us thank the Referee for providing us with useful and constructive remarks. Please find our replies below.

The Referee writes:

The authors present the obtained results for the quantum many-body scars as-is, without any exploration on how surprising their results actually are. For example, is it clear that the overlap of the quantum many-body scars with optimized Gaussian states is significantly different from that of other eigenstates in the spectrum? If not, is there some properties of the optimized Hamiltonian that change for non-quantum many-body scars?

Our response: We have implemented this suggestion in the revised version. Our newly added results show that our method works significantly better for quantum many-body scars compared to the thermal eigenstates.

The Referee writes:

The ground state of the PXP model and its quantum many-body scars close to the edges of the spectrum are known to be well-approximated by Matrix Product States (arXiv:1903.10517). This property itself makes it plausible that the quantum many-body scar states can admit several approximate parent Hamiltonians. Is this property related to the optimized non-interacting Hamiltonians that the authors discover for the quantum many-body scars close to the edge of the spectrum? For example, do the exact MPS studied in the above paper admit such non-interacting parent Hamiltonians?

Our response: We thank the Referee for this interesting question. Indeed, since the ground state of the PXP model has a large overlap with the matrix product state constructed in the reference and our Gaussian states, it follows that the constructed matrix product state can also be well-approximated by a Gaussian state, and admits a non-interacting parent Hamiltonian.

The Referee writes:

The algorithm for optimization is used as a ``black box" and no details or motivations are presented. For example, what is the role of the initial states $|\psi_\text{init} \rangle$ that dictates the initial guesses for the matrix $A$ and $B$? Do answers change for different choices of this initial state?

Our response: In the revised version, we have expanded our discussion on the choice and details of the optimization procedure, as well as the sensitivity of the results on the initial state.

The Referee writes:

Another question that the authors may consider exploring if computationally feasible: Since the ground state of the free-fermion ground state does not naturally contain the Rydberg blockade constraint, do the overlaps with the quantum many-body scars change if the many-body wavefunctions are projected and normalized within the constrained subspace?

Our response: The Referee is right that the non-interacting Hamiltonian is constructed in the full Hilbert space and therefore not all of its eigenstates can lie within the projected space. However, this does not rule out that its ground state (or even more eigenstates) can lie within this space, similarly to the Néel state. In fact, given the high overlap of the Gaussian state with the ground state, even without the projection, we know that this is the case. We have added a comment on this in the revised version.

The Referee writes:

The results on quantum many-body scars eigenstates should be contrasted with results for non-quantum many-body scar eigenstates. Perhaps the same algorithms can be applied to non-quantum many-body scars, and properties that distinguish the quantum many-body scars can be pointed out clearly.

Our response: We have implemented this suggestion in the revised version.

The Referee writes:

Potential connections to previous work addressing similar questions (arXiv:1903.10517) should be discussed clearly.

Our response: We thank the Referee for pointing out this work. We have implemented this suggestion in the revised version (please also see our response to the first point of Referee 1). We have added a reference to the reference in the revised version.

The Referee writes:

It would be good to have an overview of important details of the algorithm used.

Our response: We have implemented this suggestion in the revised version.

---

## Round 2 · Referee Report · Anonymous (Referee 1) · 2024-2-1

Strengths

The authors propose a new type of fermionic trial wavefunction approximating the scar states of the PXP model. It suggests that after a Jordan Wigner transformation these scar wavefunctions are reasonably well approximated by a Gaussian structure. This may reveal a certain property which was hitherto not pointed out.

Weaknesses

1) the fermionic quadratic Hamiltonian is different for every scar wavefunction. It is thus unclear whether there is any parent Hamiltonian that underlies the structure of all scar wavefunctions.

2) The Jordan Wigner transformation produces a non-local fermionic Hamiltonian. Logically it is not very clear why it is a good thing to make such a step.The scar states of that non-local fermionic Hamiltonian are then approximated by local quadratic BdG Hamiltonians.

3) The quadratic approximations have reasonable, but not extraordinarily good overlap with the scar wavefunctions from exact diagonalization. It is unclear to what extent these trial functions are better or conceptually more satisfactory than bosonic wavefunctions that have been proposed in the literature.

Report

The authors have clarified significantly their approach (even though several details should still be made clearer, as detailed below). Now it is possible to follow what has been done and what the results of the study imply. Below we list a few points that need to be addressed and taken into consideration.

• As the authors explain now, they apply a Jordan Wigner transformation to the Hamiltonian, to make it fermionic. If they indeed used the standard transformation of Eqs. (2-4) the resulting fermionic Hamiltonian is, however, non-local, each term containing an uncompensated tail stemming from the single s_x operator. Given that there seems to be no reason to prefer open over periodic boundary conditions.
It is important to state both facts explicitly. Also, please comment on the origin of the far off-diagonal entries in the optimal coupling matrix, and how that could come about if indeed open boundary conditions were used (see next point).

• For each scar state the authors now consider quadratic fermionic Hamiltonians. They take their ground state (which is Gaussian) and construct the inversion-symmetric cat state of Eq. (8) (which is almost surely not Gaussian, as should be stated if indeed true). The overlap of these cat states with a considered scar state is the maximized by adjusting the Hamiltonian. The authors claim that the optimal Hamiltonians are local. However, Fig. 3 shows very strong matrix elements far off the diagonal. This would be natural to expect if periodic boundary conditions had been used, but is very unnatural for open boundary conditions, and seems to directly contradict the claim of locality of the fermionic Hamiltonian.
This issue has to be clarified and thoroughly discussed, possibly together with revisiting the Jordan Wigner transformation.

• The main aim of this work was to elucidate the origin of many-body scars. However, it remains unclear what insight has been gained. In the introduction, the authors write: “Indications for a Gaussian structure of quantum many-body scars would suggest that the PXP model is close to certain quadratic parent Hamiltonians, hinting on the origin of quantum many-body scars”. However, the meaning of “certain parent Hamiltonians” is unclear. Namely, as is stated in the conclusion, each individual scar state has been approximated by the ground state of a different quadratic Hamiltonian.
To justify the notion of a “parent Hamiltonian”, the authors should at least offer a speculation regarding the relation among the different quadratic Hamiltonians associated with all the scar states (are they close to each other in some sense? Or are they all representatives of some sub-class of Hamiltonians)?

• Judging from the numerical quality of the overlap, the fermionic non-interacting trial wavefunctions presented here do not seem as good as the wavefunctions constructed from (non-interacting) bosonic magnon excitations above certain reference states in the literature. Insofar it is not clear whether any statement as to a hidden fermionic (as opposed to bosonic) nature of QMBS can be made.
A statement about this consideration would help the reader situating this approach within the landscape of others.

• One detail in the text: On p. 3, 2nd column, the meaning of the following sentences is unclear: “One could alternatively choose A and B such that the ground state of Hˆ is given by the product state that has the highest overlap with the quantum many-body scar under consideration.” Do the authors mean the initial choice of A and B? They seem to exactly describe what was done anyhow, and it is thus unclear what the alternative is supposed to be.
Please clarify.
• There are still many typos and English mistakes that remain to be corrected.

In conclusion: It has become much clearer what was done in this work. With the above clarifications, the reader will have enough information to form a well-informed opinion about this fermionic approach to PXP.

With these changes the paper might just make the bar for publication in SciPost.
However, we still feel that none of the expected acceptance criteria is met.
At best the manuscript approaches criterion 4: “Provide a novel and synergetic link between different research areas.”
This may hold if one interprets different approaches to the PXP model as different research areas. The analysis is novel. Whether the link is synergetic is not so clear, though.

Requested changes

1) State that the JW transformation yields a non-local Hamiltonian, and that it makes no difference whether one consideres open or periodic boundary conditions. In fact periodic boundary conditions would be more natural, as translation symmetry would not be broken.

2) Please comment on the origin of the far off-diagonal entries in the optimal coupling matrix, and how that could come about if indeed open boundary conditions were used. This issue has to be clarified and thoroughly discussed, possibly together with revisiting the Jordan Wigner transformation.

3) To justify the notion of a “parent Hamiltonian”, the authors should at least offer a speculation regarding the relation among the different quadratic Hamiltonians associated with all the scar states (are they close to each other in some sense? Or are they all representatives of some sub-class of Hamiltonians)?

4) Comment on the quality of the approximation by fermionic wavefunctions, as compared to that with bosonic trial wavefunctions.

  • validity: good
  • significance: low
  • originality: ok
  • clarity: high
  • formatting: excellent
  • grammar: good

Author:  Wouter Buijsman  on 2024-07-01  [id 4595]

(in reply to Report 1 on 2024-02-01)

Let us thank the referee again for providing us a positive and constructive report. Please find our reply to each of the points below.

The referee writes:

The fermionic quadratic Hamiltonian is different for every scar wavefunction. It is thus unclear whether there is any parent Hamiltonian that underlies the structure of all scar wavefunctions.

Our response:

The primary aim of our work is to show that individual quantum many-body scars can be described surprisingly well in terms of Gaussian states. For such states, the number of parameters scales polynomially with system size, instead of exponentially (as for generic quantum many-body states). We believe that such an observation is non-trivial. Our work should be seen as a first step in exploring the posssibility that a unified description in terms of Gaussian states for all quantum many-body scars can be obtained. We have commented on this more explicitly in the revised version.

The referee writes:

The Jordan-Wigner transformation produces a non-local fermionic Hamiltonian. Logically it is not very clear why it is a good thing to make such a step. The scar states of that non-local fermionic Hamiltonian are then approximated by local quadratic BdG Hamiltonians.

Our response:

While the fermionic version of the PXP Hamiltonian is indeed non-local, we do not use it in any way in the manuscript, since we work directly with its eigenstates. Since the quantum many-body scars are known to have decaying spatial correlations, and sub-volume entanglement entropy it is actually natural that the optimal quadratic Hamiltonian is local. With this said, we do not constrain the approximating quadratic Hamiltonian, such that the outcome of the optimization could have been non-local. We have clarified this issue in the text.

The referee writes:

The quadratic approximations have reasonable, but not extraordinarily good overlap with the scar wavefunctions from exact diagonalization. It is unclear to what extent these trial functions are better or conceptually more satisfactory than bosonic wavefunctions that have been proposed in the literature.

Our response:

Our approach is conceptually different from what has been done in previous works (please also see our previous reply). Our goal is not to achieve the most accurate description of quantum many-body scars. Instead, we aim to find out whether quantum many-body scars can be well described in terms of Gaussian states. We have elaborated more extensively on this in the revised version.

The referee writes:

As the authors explain now, they apply a Jordan-Wigner transformation to the Hamiltonian, to make it fermionic. If they indeed used the standard transformation of Eqs. (2-4) the resulting fermionic Hamiltonian is, however, non-local, each term containing an uncompensated tail stemming from the single $\sigma^x$-operator. Given that there seems to be no reason to prefer open over periodic boundary conditions. It is important to state both facts explicitly.

Our response:

We thank the referee for this comment, which allowed us to considerably improve our manuscript. We have changed the calculation to periodic boundary conditions, which allowed us to obtain remarkable overlaps also with scars close to the center of the band, compared to the previous open boundary conditions calclations. We have added a comment about the non-locality of the Jordan-Wigner transformed Hamiltonian to the text.

The referee writes:

Also, please comment on the origin of the far off-diagonal entries in the optimal coupling matrix, and how that could come about if indeed open boundary conditions were used (see next point).

Our response:

We thank the referee for making this observation. In the revised version, we focus periodic instead of open boundary conditions, for which this issue does not appear. Switching from open to periodic boundary conditions does not qualitatively change our findings or conclusions. In fact, we find that overlaps are typically higher compared to what we observed before. In particular, the overlaps improved significantly for the quantum many-body scars near the center of the spectrum.

The referee writes:

For each scar state the authors now consider quadratic fermionic Hamiltonians. They take their ground state (which is Gaussian) and construct the inversion-symmetric cat state of Eq. (8) (which is almost surely not Gaussian, as should be stated if indeed true). The overlap of these cat states with a considered scar state is the maximized by adjusting the Hamiltonian.

Our response:

The symmetrized cat state are indeed not Gaussian. We have mentioned this explicitly in the revised version of the manuscript. Let us mention that we perform optimizations for non-symmetrized (Gaussian) states in appendix A.

The referee writes:

The authors claim that the optimal Hamiltonians are local. However, Fig. 3 shows very strong matrix elements far off the diagonal. This would be natural to expect if periodic boundary conditions had been used, but is very unnatural for open boundary conditions, and seems to directly contradict the claim of locality of the fermionic Hamiltonian. This issue has to be clarified and thoroughly discussed, possibly together with revisiting the Jordan Wigner transformation.

Our response:

In the revised version, we focus on periodic instead of open boundary conditions. For periodic boundary conditions, the issue does not appear.

The referee writes:

The main aim of this work was to elucidate the origin of many-body scars. However, it remains unclear what insight has been gained. In the introduction, the authors write: “Indications for a Gaussian structure of quantum many-body scars would suggest that the PXP model is close to certain quadratic parent Hamiltonians, hinting on the origin of quantum many-body scars”. However, the meaning of “certain parent Hamiltonians” is unclear. Namely, as is stated in the conclusion, each individual scar state has been approximated by the ground state of a different quadratic Hamiltonian.

To justify the notion of a “parent Hamiltonian”, the authors should at least offer a speculation regarding the relation among the different quadratic Hamiltonians associated with all the scar states (are they close to each other in some sense? Or are they all representatives of some sub-class of Hamiltonians)?

Our response:

We thank the referee for pointing out this omission. As elaborated above, our main result is the obervation that quantum many-body scars can be well described in terms of Gaussian states, the implications on the structure of models hosting quantum many-body scars are merely speculations that can provide starting points for future investigations. We have stressed this more explicitly in the revised version, and expanded the discussion on how different parent Hamiltonians could possible be related in the conclusions and outlook section.

The referee writes:

Judging from the numerical quality of the overlap, the fermionic non-interacting trial wavefunctions presented here do not seem as good as the wavefunctions constructed from (non-interacting) bosonic magnon excitations above certain reference states in the literature. Insofar it is not clear whether any statement as to a hidden fermionic (as opposed to bosonic) nature of QMBS can be made. A statement about this consideration would help the reader situating this approach within the landscape of others.

Our response:

We have implemented this suggestion in the revised version. Specifically, we have added a comment similar to the one made by the referee in the conclusions and outlook section.

The referee writes:

One detail in the text: On p. 3, 2nd column, the meaning of the following sentences is unclear: “One could alternatively choose $A$ and $B$ such that the ground state of $\hat{H}$ is given by the product state that has the highest overlap with the quantum many-body scar under consideration.” Do the authors mean the initial choice of $A$ and $B$? They seem to exactly describe what was done anyhow, and it is thus unclear what the alternative is supposed to be. Please clarify.

Our response:

The referee is correct regarding the first point. In the revised version, we have replaced “choose $A$ and $B$" by “choose the initial guesses for $A$ and $B$". Please let us stress that this is not the procedure adapted in the manuscript: we take $A$ and $B$ such that the ground state of $\hat{H}$ is given by the $\mathbb{Z}_2$ state. Although for some quantum many-body scars the highest-overlapping basis state is the $\mathbb{Z}_2$ state, this is not the case in general.

The referee writes:

There are still many typos and English mistakes that remain to be correct

Our response:

We have carefully checked the manuscript, and corrected all misprints that we found.

---

## Round 2 · Author Response

Dear Editor,

We thank the Referees for their useful comments, that helped us to improve the manuscript. Please find below our replies to each of the points raised in the reports. We hope you find the revised version of the manuscript suitable for publication in SciPost Physics.

Yours sincerely,
Wouter Buijsman
Yevgeny Bar Lev

---

## Round 2 · List of Changes

• Elaborated on the motivation for our work.

  • Elaborated on the mapping of the PXP model to the fermion model.

  • Elaborated more on the interpretation of the numerical results.

  • Added an Appendix on the performance of the optimization algorithm on thermal (non-scarred) eigenstates.

  • Implemented minor corrections on issues pointed out in the reports.

---

## Round 3 · Referee Report · Anonymous (Referee 1) · 2024-7-16

Strengths

As previously described

Weaknesses

As previously described

Report

Our questions have been addressed.
Now the authors have switched to analyzing the model with periodic boundary conditions and seem to obtain cleaner structures for their wavefunctions.

The point (site 1) where the Jordan Wigner transformation is rooted breaks translational invariance of the resulting fermionic Hamiltonian. This is (at least) reflected in sign changes of the elements of A and B matrices as the root is crossed. This should be discussed.

The resulting A and B matrices are surprisingly invariant upon translation by 2 sites. However, the eigenstates of the original Hamiltonian are obviously translation invariant (by one site). To what extent are the quadratic trial wavefunctions close to being 1-site translation invariant, even though they do not have explicitly this structure? This should be commented on.

Now the procedure and approach are explained clearly, such that a reader can appreciate what the content of the work is.
I have commented previously on the limited scope of the approach. This has remained unchanged and the work just barely makes the bar of some of the criteria for SciPost. However, the work can be published.

Requested changes

  1. The degree to which trial scar states are approximately translationally invariant should be discussed, as well as the reason, why translation invariance is not enforced by applying a symmetrizer.

  2. The effect of the choice of the root for the JW transform should be mentioned, as it is indeed reflected in the structure of A and B.

Recommendation

Ask for minor revision

  • validity: ok
  • significance: low
  • originality: ok
  • clarity: good
  • formatting: excellent
  • grammar: good

Author:  Wouter Buijsman  on 2024-07-31  [id 4668]

(in reply to Report 1 on 2024-07-16)

We thank the Referee for their constructive report and positive attitude towards our manuscript. Please find our reply to both of the points below.

The referee writes:

The point (site 1) where the Jordan-Wigner transformation is rooted breaks translational invariance of the resulting fermionic Hamiltonian. This is (at least) reflected in sign changes of the elements of $A$ and $B$ matrices as the root is crossed. This should be discussed.

Our response:

The remark made by the Referee is correct, and is in agreement with what we observe in the color plots of matrices $A$ and $B$ as shown in Fig. 3. We have added this remark in the revised version.

The referee writes:

The resulting $A$ and $B$ matrices are surprisingly invariant upon translation by 2 sites. However, the eigenstates of the original Hamiltonian are obviously translation invariant (by one site). To what extent are the quadratic trial wavefunctions close to being 1-site translation invariant, even though they do not have explicitly this structure? This should be commented on.

Our response:

Let us remind the Referee that we focus on the overlap of symmetrized Gaussian states [defined in Eq. (7)] with quantum many-body scars. The symmetrization of a state invariant by translation over two sites leads to a state translationally invariant by one site. With this in mind, it does not come as a surprise that the resulting $A$ and $B$ matrices are almost invariant upon translation by two sites. If the overlap between a (symmetrized) trial wavefunction and a quantum many-body scar is close to unity, the trial wavefunction and the quantum many-body scar are nearly identical. Then, the trial wavefunction is almost translationally invariant. We have elaborated on both of these points in the revised version.

---

## Round 3 · Author Response

Dear Editor,

We are grateful to the Referee for providing us with useful comments, remarks and suggestions. We have implemented all requested changes in the revised version. Please find below our reply to each of the points in the report. Hereby, we would like to resubmit our manuscript to SciPost Physics.

Yours sincerely,
Wouter Buijsman
Yevgeny Bar Lev

---

## Round 3 · List of Changes

• Changed from open to periodic boundary conditions.

  • Clarified on the implementation of the Jordan-Wigner transformation.

  • Expanded the discussion on possible properties of parent Hamiltonians.

  • Commented on the quality of the approximations in view of previous results.

  • Implemented minor corrections on issues pointed out in the report.

---

## Round 4 · Author Response

Dear Editor,

We are grateful to the Referee for their constructive comments and recommendation to accept our manuscript for publication in SciPost Physics. Please find below our reply to both points in the report.

Yours sincerely,
Wouter Buijsman
Yevgeny Bar Lev

---

## Round 4 · List of Changes

- Expanded the discussion of the Jordan-Wigner transform with a remark on the choice of the root.

-Commented on the translational invariance of the trial wavefunctions.

---

## Editorial Decision

published